# Characteristics of NDVI Changes in the Altay Region from 1981 to 2018 and Their Relationship to Climatic Factors

**Yang Yan** [1,2] 🄳, **Junhui Cheng** [1,2], **Yongkang Li** [1,2] 🄳, **Jie Fan** [1,2] and **Hongqi Wu** [1,2,*]

1   College of Resources and Environment, Xinjiang Agricultural University, Urumqi 830052, China
2   Xinjiang Key Laboratory of Soil and Plant Ecological Processes, Xinjiang Agricultural University, Urumqi 830052, China
*   Correspondence: whq@xjau.edu.cn

**Abstract:** Vegetation growth and its response to climatic factors have become one of the most pressing issues in ecological research. However, no consensus has yet been reached on how to resolve this problem in arid areas with a high-elevation gradient and complex underlying surface. Here, NOAA CDR AVHRR NDVI V5 for 1981–2018 and China's regional surface meteorological faction-driven datasets were used. General linear regression, the Mann-Kendall test and sliding *t*-test, Pearson correlations, and the Akaike information criterion (AIC), on a grid-scale, were applied to analyze the annual normalized difference vegetation index (NDVI) and its relationship with temperature and precipitation in the Altay region. Results revealed that the temporal trend of NDVI for most grid cells was non-significant. However, mountains, coniferous forests, grasslands, and meadows in the high-elevation zone displayed a slow increasing trend in NDVI. Further, NDVI was positively correlated with the mean annual temperature and total annual precipitation, the latter playing a more significant role. Yet, for desert and shrub vegetation and coniferous forest, their NDVI had insignificant negative correlations with the mean annual temperature. Hence, both the trends and drivers of NDVI of high elevation are highly complex. This study's findings provide a reference for research on vegetation responses to climate change in arid areas having a high-elevation gradients and complex underlying surfaces.

**Keywords:** NDVI; complex underlying surface; high elevations; spatial and temporal variation characteristics; climate-driven; Altay region

## 1. Introduction

As the main biotic component of terrestrial ecosystems [1,2], vegetation plays an irreplaceable role in carbon balance regulation, soil and water conservation, and the maintenance of ecosystem stability [3–5]. Vegetation growth is sensitive to climate change [6,7]. Accordingly, it is of great practical significance to study the temporal and spatial characteristics of vegetation growth and its response to climate change under the background of global climate change [8]. A normalized difference vegetation index (NDVI) can accurately convey the information of vegetation cover and growth status, and is not susceptible to influences from cloud cover, terrain, solar elevation angle, and other factors. For these reasons, NDVI is widely used to study the characteristics of vegetation change under long time series [9–11].

The emergence of long time series NDVI datasets, such as AVHRR NDVI, GIMMS NDVI, MODIS NDVI, SPOT-VGT, and so on, made it possible to study the change trend of vegetation under long time series [12,13]. Numerous studies worldwide have investigated the interannual variation of vegetation growth and its driving forces at different spatio–temporal scales. However, due to the influence of the studied area, and the scale and methods used, their findings are not consistent. For example, Liu et al. [14] used the Theil–Sen median slope method to analyze the annual global NDVI from 1982 to 2012;

they found that NDVI of most regions showed an increasing–decreasing–increasing trend across years, for which the year of abrupt change were 1995 and 2004. Li et al. [15] applied the same method, finding that NDVI of Central Asia showed an increasing trend during 1982–1998, but a decreasing trend during 1998–2013. Yao et al. [16] used the Mann-Kendall trend analysis method to test NDVI of Xinjiang during 1981–2010, showing that the NDVI trend significantly increased and then decreased, and the year of abrupt change also occurred in 1998. In addition to scale effect, different vegetation types and responses to climate change in different areas may also be the reasons for the different trends of NDVI and the year of abrupt change. Gouveia et al. [17] used NDVI to study the response of five vegetation types to climate change in the Mediterranean; they found that temperate marine vegetation responded most to drought intensity, in that as the drought intensity increased, the vegetation's NDVI declined rapidly. Bao et al. [18] analyzed vegetation NDVI of Mongolia during 1982–2010, reporting that NDVI stagnated or even declined during 1991–1994, being most pronounced in meadow, grassland, and forest areas. Li et al. [19] used the second-order correlation to analyze the response of different vegetation types of NDVI to temperature in the North–South transition zone of China during 1982–2015, and found that only the NDVI of evergreen broad-leaved forest was rising, while others were declining. These studies found that the inter-annual trends of NDVI of different vegetation types in the same study area were different, and the same vegetations would show different changes in different study areas. In addition, they respond differently to meteorological factors, where there is a strong correlation between vegetation change and temperature or precipitation. Moreover, such correlations can also significantly differ among various regions [20,21]. Based on MODIS NDVI data and ERA5 meteorological data, Wei et al. [22] calculated the risk-detector and found that the influence of temperature was most significant in Northeast, East and central China during 2001–2020, while the influence of precipitation was most significant in North, South and Northwest China. Work by Song et al. [23] uncovered, by analyzing NDVI from 1982 to 2005, that total annual precipitation had the greatest impact on it in arid and semi-arid areas, while annual mean temperature had differential effects on vegetation. Yang et al. [24] studied the trends and factors influencing of NDVI during the growing season of vegetation in five climatic regions of the Qinghai–Tibet Plateau, detecting significant differences in the dominant roles of air temperature and precipitation in differing climatic regions, but the effect of overall air temperature on NDVI was generally stronger than that of precipitation. Yu et al. [25] used Pearson correlations to analyze the relationships between NDVI and climate factors in Xinjiang, showing that precipitation has a greater impact on NDVI of this region, being more significant in northern Xinjiang. Despite the mounting studies of vegetation change, the research scale is often too large, the response of vegetation to climate is too general, and there is a dearth of specific studies focused on different underlying surface types. In addition, consistent conclusions remain elusive from corresponding studies on arid areas featuring a high-elevation gradient and complex underlying surface.

The Altay region is located in the arid hinterland of Eurasia continent and its underlying surface is complex and diverse, and the ecological environment is fragile. Elevation in the Altay region spans more than 4000 m, where the vegetation's vertical zonality is stark. In the context of ongoing global climate change, the climate in this region has undergone obvious "warming and humidification" changes [26,27]. Therefore, it is an ideal, natural test site for studying vegetation growth changes in arid areas with a high-elevation gradient and complex underlying surface. In addition, the Altay region is a necessary passage of the Silk Road, connecting many countries in Central Asia. The ecological problems caused by the changing ecological environment are also important factors affecting the political and economic stability of this region. The data source for this paper is NOAA CDR AVHRR NDVI V5. We used grid cells to study the spatial and temporal variation trends and driving factors of vegetation's NDVI of the Altay region in a recent 38-year period, across a suite of geomorphic and vegetation types. Our findings provide scientific theoretical support for the protection and utilization of vegetation in this kind of fragile ecological environment.

## 2. Materials and Methods

### 2.1. Study Area

The Altay region lies in the northern part of Xinjiang Uygur Autonomous Region at these geographical coordinates: 85°31′36″–91°01′15″ E, 44°59′35″–49°10′45″ N. It is located in the southwest foot of the middle portion of the Altay Mountains and north of the Junger Basin. According to the geomorphic types, the region can be divided into four geomorphic units: northern mountain area, central platform, central plain valley area, and southern hilly area. These accounting for 32.3%, 10.2%, 27.0%, and 30.5% of the region's total area, respectively. The region has a temperate continental climate. Annual precipitation is 400–600 mm in the northern mountains, 150–200 mm in the central plains and tablelands, and about 95 mm in the southern hills. Overall, the region's annual mean temperature is 3.7 °C, but it can vary by up to 30 °C; in mountainous and hilly areas it is lower than 4 °C, while in plains and platform areas it is higher than 4 °C [28]. Affected by terrain and other factors, the northern mountains are mainly covered with alpine vegetation, coniferous forests, meadows, and shrubs. Grassland is mainly distributed at lower elevation in the foothills. Broad-leaved forest and cultivated vegetation are more widely distributed along the region's rivers. Desert is mostly found in the southern part of the region at low elevation, where it is relatively dry [29]. Generally, the vegetation in the Altay region is rich and diverse, and seriously affected by global climate change [30].

### 2.2. Data Collection and Processing

#### 2.2.1. NDVI Dataset

This paper used the NOAA CDR AVHRR NDVI V5 at a 5000 m spatial resolution, for 1981–2018, provided by the National Oceanic and Atmospheric Administration. These datasets have been widely used to study changes to NDVI under long time series, because of its long time series (1981—current day) and medium spatial resolution [31,32]. Although there are many NDVI product datasets with a higher resolution, they are not optimal in this study due to the shorter time series they provide [33].

The maximum value of annual NDVI is an important phenological attribute, which can make the data dimension decrease exponentially [34]. The online platform Google Earth Engine (GEE) is a high-performance cloud-based platform that not only has access to a large and growing amount of earth observation data, but also the ability to analyze and process data [11,35]. Therefore, a total number of 4892 NDVI grid cells of the Altay region were extracted by annual maximum synthesis and masking on the GEE.

#### 2.2.2. Climate Dataset

Climate data were derived from the 0.1° resolution China Meteorological Forcing Dataset (CMFD), provided by the Tibetan Plateau Data Center [36]. This dataset was based on the existing Princeton reanalysis data, GLDAS data, GEWEX-SRB radiation data and TRMM precipitation data, and integrates the routine meteorological observation data of the China Meteorological Administration [37,38]. More stations were used to generate CMFD and the accuracy of superior quality, than the existing international reanalysis data. Due to its continuous temporal coverage and consistent quality, CMFD has become one of the most widely used climate datasets in China. The datasets not only show strong adaptability in the complex underlying surface in high elevation, but are also representative of the arid area of Xinjiang [39].

Because the spatial resolution of CMFD dataset is low, there may be errors in the analysis based on grid cells. To this end, we used a geographically weighted regression (GWR) model to measure the annual mean temperature (K) and annual mean precipitation rate (mm·hr$^{-1}$) from 1981 to 2018, downscaling to 5000 m spatial resolution [40–42]. The annual mean precipitation rate was converted to total annual precipitation (mm). Then the climate data was applied a spatio–temporal and NDVI dynamic analyses.

### 2.2.3. Geomorphic and Vegetation Regionalization Data

Geomorphic regionalization data were selected from the 1:1,000,000 spatial distribution dataset of geomorphic types, provided by the Cloud Platform of Resources and Environmental Data, Chinese Academy of Sciences [43]. This study used four landform types: mountains, platforms, plains, and hills in the Altay region (Figure 1b) for the analysis.

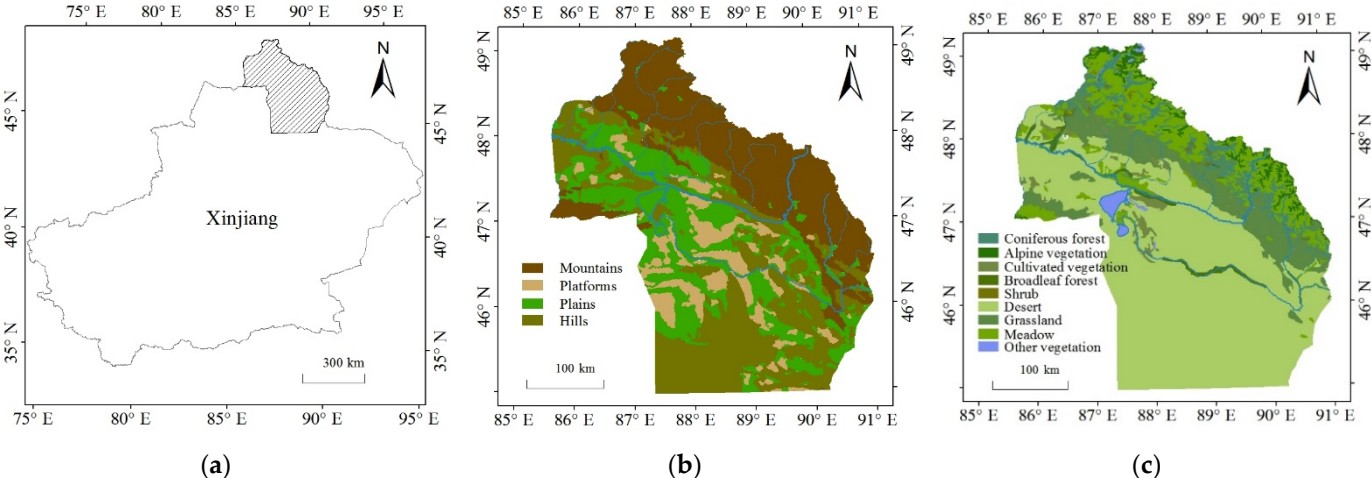

**Figure 1.** Summary maps of the study area: (**a**) geographical location and distribution of (**b**) its geomorphic types and (**c**) vegetation types.

Vegetation regionalization data were selected from the 1:1,000,000 Vegetation Map of China, provided by the Cloud Platform of Resources and Environmental Data, Chinese Academy of Sciences [44]. Nine vegetation types, coniferous forest, alpine vegetation, cultivated vegetation, broadleaf forest, shrub, desert, grassland, meadow, and 'other vegetation' (Figure 1c), were extracted for the analysis.

These data were published in 2001 and 2009, respectively. Although there may be some changes in the current vegetation distribution, they are still the most authoritative data in the application of vegetation classification and geomorphic classification in China. Elevation distribution characteristics of geomorphic and vegetation types are clear. Figure 2 shows the distribution of geomorphic types and vegetation types at different elevations.

### 2.2.4. The Digital Elevation Model (DEM)

The digital elevation model (DEM) was obtained from the website of United States Geological Survey (USGS), and the complete DEM data of Altay region were obtained by stitching and clipping.

Sources of cartographic data and statistics are listed in Table 1.

**Table 1.** Sources of principal data.

| Data Name | Data Resolution | Data Source |
|---|---|---|
| NDVI dataset | 5000 m (yearly) | National Oceanic and Atmospheric Administration (http://www.noaa.gov/web.html, accessed on 1 June 2022) |
| Climate dataset | 10,000 m (yearly) | National Data Center for Tibetan Plateau Science (http://data.tpdc.ac.cn, accessed on 27 July 2022) |
| Vegetation regionalization data | 1:1,000,000 | Cloud Platform of Resources and Environmental Data, Chinese Academy of Sciences (http://www.resdc.cn, accessed on 10 September 2022) |
| Geomorphic regionalization data | 1:1,000,000 | Cloud Platform of Resources and Environmental Data, Chinese Academy of Sciences (http://www.resdc.cn, accessed on 10 September 2022) |
| SRTM | 90 m | CGIAR Consortium for Spatial Information (http://srtm.csi.cgiar.org/, accessed on 1 June 2022) |

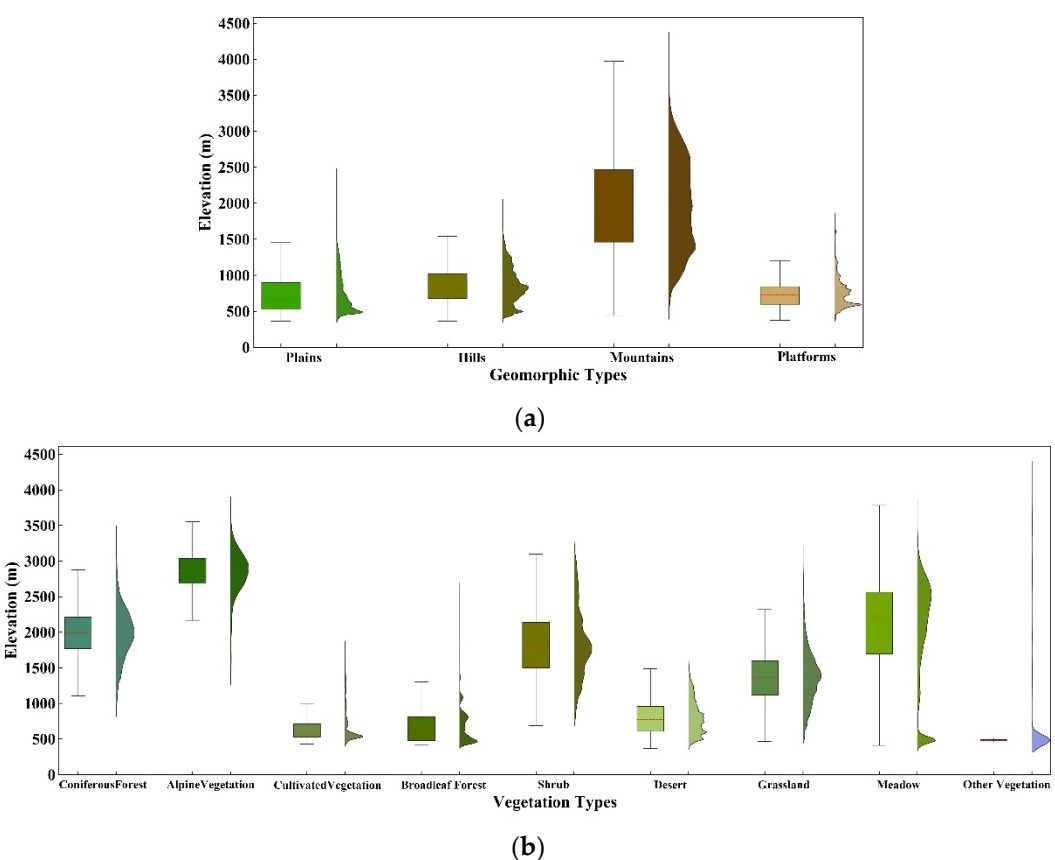

**(a)**

**(b)**

**Figure 2.** Altay region elevation distribution: (**a**) distribution of geomorphic types, (**b**) distribution of vegetation types.

### 2.3. Methods

First, we used the Sen's trend slope and a quadratic polynomial equation model to analyze the interannual trends of the annual maximum NDVI on each grid cell of its own study period, separately [45,46]. In order to fully capture the trends of NDVI, corresponding linear models (increasing and decreasing) and nonlinear models (increasing and then decreasing unimodal trend or decreasing and then increasing U-shaped trend) were fitted in parallel. Among them, the most representative model was selected to capture the spatial distribution characteristics of the trend in NDVI change in the Altay region [47]. The screening principle for the optimal model was as follows: when only one of the linear or nonlinear relationships between NDVI and year was significant, this relationship was used as the optimal fitting model of that particular grid cell. When both linear and nonlinear models fit well, the optimal fitted model was selected according to the Akaike information criteria (AIC) (the lower the AIC value, the better the model fitting effect) [48]. When both the linear and nonlinear relationships of the maximum yearly NDVI trends were not significant, this proved that the relationship between the two was negligible [49]. Secondly, the Mann-Kendall test and sliding *t*-test were further used to determine the corresponding abrupt-change year of NDVI [50]. Finally, we used a Pearson's correlation coefficient to analyze each relationship between NDVI and climate factors in the Altay region [51]. All the above analyses were completed in the R software. In order to understand the methods we used in this paper, the analysis workflow chart is shown in Figure 3.

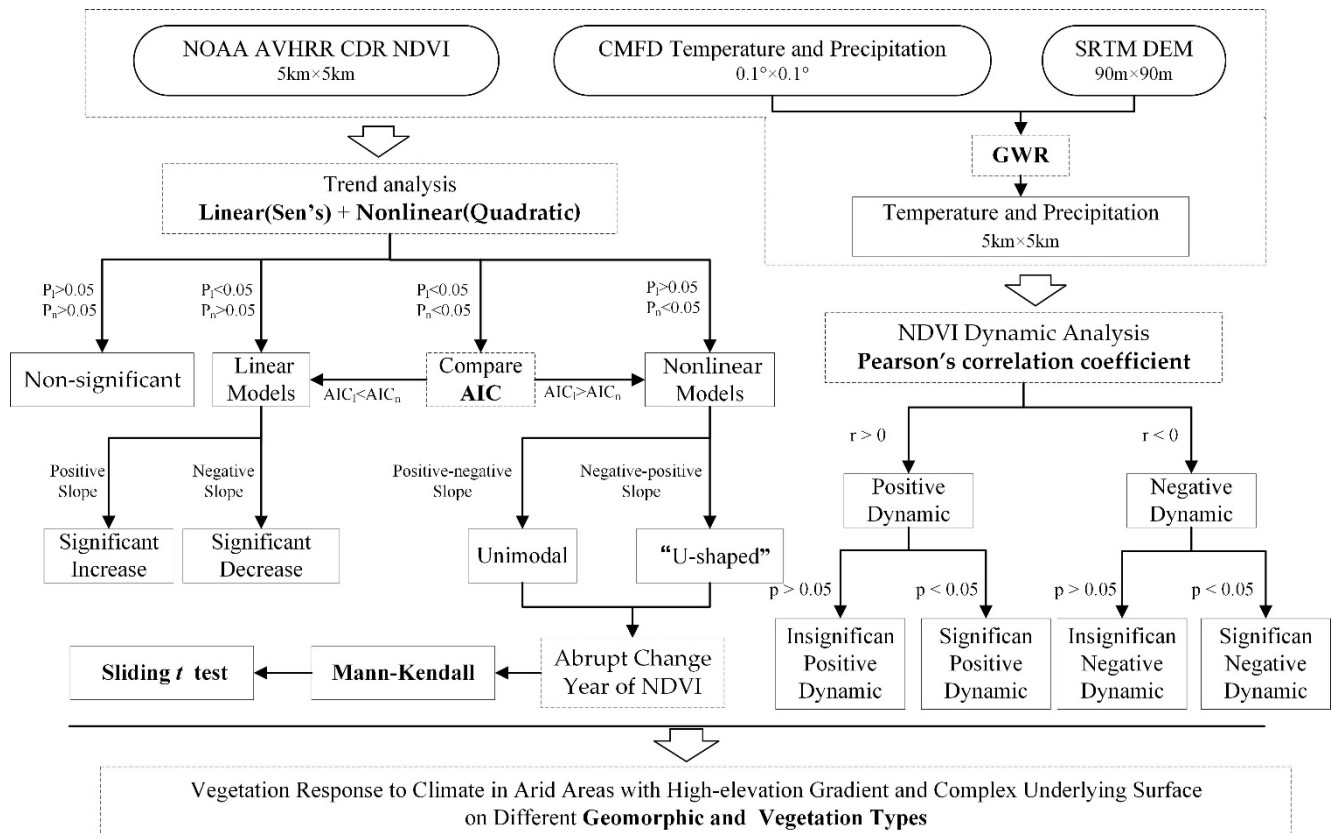

**Figure 3.** Analysis workflow chart.

## 3. Results

### 3.1. Interannual Variation in NDVI

The fitted general linear model revealed that the trends of NDVI over the past 38 years (1981–2018) in the Altay region could be classified into five categories. Specifically, 55.6% of the grid cells in the region displayed no significant relationship between NDVI and year ($p > 0.05$) (Table 2). In the remaining 44.4% of grid cells, there were four types of NDVI trends in the past 38 years: significant increase, significant decrease, unimodal (first increasing, then decreasing), and "U-shaped" (first decreasing, then increasing) ($p < 0.05$) (Table 2). Of those four, a significant increase trend was prevalent, accounting for 35.2% of the number of grid cells in the study area (Table 2). Significant reduction, unimodal and U-shaped trends together accounted for 9.2% of the total number of grid cells (Table 2). Additionally, from the distribution trend of NDVI, we found that the grid cells with non-significant trends were distributed around south of the Altay region, and the unimodal and "U-shaped" grid cells were more scattered (Figure 4).

**Table 2.** Interannual variation of NDVI of geomorphic units in the Altay region from 1981 to 2018.

| Geomorphic Type | Significant Increase | Significant Decrease | Unimodal | "U-Shaped" | Non-Significant |
|---|---|---|---|---|---|
| All | 1722 (35.2%) | 50 (1.0%) | 357 (7.3%) | 45 (0.9%) | 2718 (55.6%) |
| Plains | 447 (33.8%) | 11 (0.8%) | 91 (6.8%) | 26 (1.9%) | 745 (56.4%) |
| Platforms | 135 (27.0%) | 4 (0.8%) | 16 (3.2%) | 14 (2.8%) | 330 (66.1%) |
| Hills | 378 (25.3%) | 9 (0.6%) | 82 (5.5%) | 5 (0.3%) | 1016 (68.1%) |
| Mountains | 762 (48.1%) | 26 (1.6%) | 168 (10.6%) | 0 (0.0%) | 627 (39.6%) |

Values are the number of grid cells (proportion) of the NDVI of different change characteristics in each landform type.

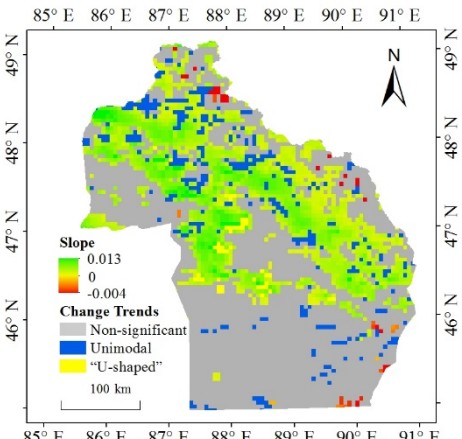

**Figure 4.** Spatial distribution of changes in the NDVI of the Altay region, from 1981 to 2018. From the perspective of geomorphic types, the trend for NDVI was of increasing significantly mainly in mountainous areas (48.1% of all grid cells). For the other three landforms, the majority of their changes in NDVI consisted of non-significant trends, for which the proportion of grid cells was 56.45% (plains) to 68.1% (hills) (Table 2).

Among vegetation types, NDVI of coniferous forest, cultivated vegetation, broadleaf forest, grassland, and meadow mostly increased significantly, this trend accounting for 43.3% (meadow) to 75.2% (cultivated vegetation) of the total number of grid cells. However, alpine vegetation, shrub, desert and other vegetation types showed non-significant changes (Table 3). At the same time, the interannual variation pattern of the NDVI differed among vegetation types; for example, the "U-shaped" trend was not found for the distributions of coniferous forest, alpine vegetation, shrub, meadow, and other vegetation types (Table 3). A significant decreasing trend was not detected in cultivated vegetation, broad-leaved forest, and shrub distributions (Table 3). Based on the above analysis, evidently the vegetation types in the Altay region are primarily undergoing a greening trend. Under the influence of human activities, the NDVI of cultivated vegetation significantly increased to the greatest extent. In addition, the growth of NDVI was pronounced in coniferous forest, grassland, and broadleaf forest near river valleys and high-elevation areas.

**Table 3.** Interannual variation in the NDVI of different vegetation types in the Altay region, from 1981 to 2018.

| Vegetation Type | Significant Increase | Significant Decrease | Unimodal | "U-Shaped" | Non-Significant |
|---|---|---|---|---|---|
| Coniferous forest | 120 (53.0%) | 1 (0.4%) | 38 (16.8%) | 0 (0%) | 67 (29.6%) |
| Alpine vegetation | 56 (30.1%) | 4 (2.1%) | 4 (2.1%) | 0 (0%) | 122 (65.5%) |
| Cultivated vegetation | 88 (75.2%) | 0 (0%) | 11 (9.4%) | 3 (2.5%) | 15 (12.8%) |
| Broadleaf forest | 45 (55.5%) | 0 (0%) | 8 (9.8%) | 1 (1.2%) | 27 (33.3%) |
| Shrub | 8 (34.7%) | 0 (0%) | 6 (26.0%) | 0 (0%) | 9 (39.1%) |
| Desert | 662 (23.0%) | 25 (0.8%) | 142 (4.9%) | 40 (1.3%) | 2005 (69.7%) |
| Grassland | 426 (66.0%) | 1 (0.1%) | 70 (10.8%) | 1 (0.1%) | 147 (22.7%) |
| Meadow | 297 (43.3%) | 17 (2.4%) | 78 (11.3%) | 0 (0%) | 293 (42.7%) |
| Other vegetation | 20 (36.3%) | 2 (3.6%) | 0 (0%) | 0 (0%) | 33 (60.0%) |

Values are the number of grid cells (proportion) of the NDVI of different change characteristics in each vegetation type.

*3.2. NDVI ofterannual Trend in Geomorphic and Vegetation Types*

For areas with significant changes in the NDVI over the years, histograms of their slopes show that the proportion of grid cells with an increasing trend significantly exceeds those with a decreasing trend, with higher slope values for the former than the latter (Figure 5). The slope for a significant increase ranges from $0.01 \cdot 10 \, \text{y}^{-1}$ to $0.11 \cdot 10 \, \text{y}^{-1}$, but is

centered mainly at $0.03 \cdot 10 \text{ y}^{-1}$. The slope values with a downward trend were mainly at $-0.02 \cdot 10 \text{ y}^{-1}$ (Figure 5).

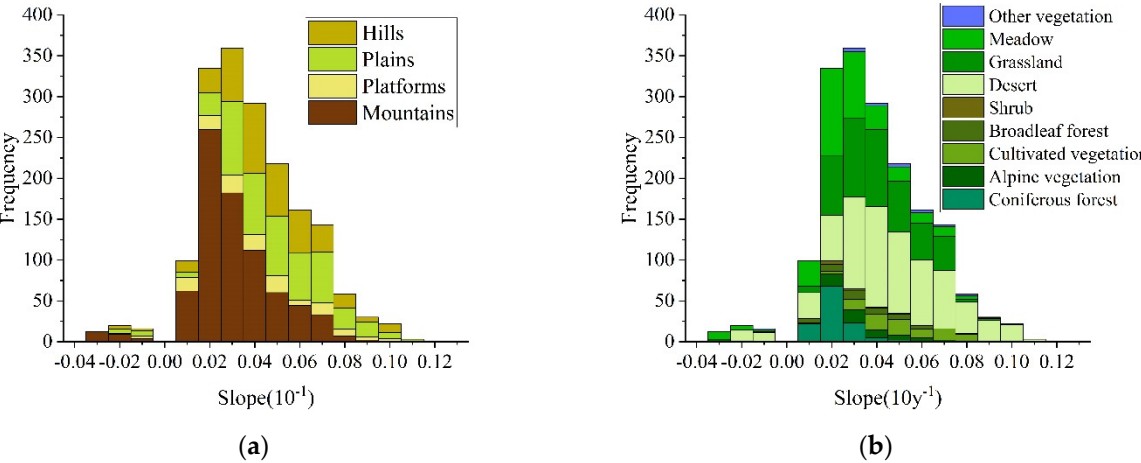

**Figure 5.** Histograms for slopes of NDVI change in the Altay region from 1981 to 2018: (**a**) significant increase and decrease in the NDVI of geomorphic types, (**b**) significant increase and decrease in the NDVI of vegetation types.

The slopes for increasing NDVI, in both hilly and platform areas, were mainly concentrated around a value of $0.04 \cdot 10 \text{ y}^{-1}$, while in plains it was $0.03 \cdot 10 \text{ y}^{-1}$, and in mountainous areas it was $0.02 \cdot 10 \text{ y}^{-1}$ (Figure 5a). The slopes for decreasing NDVI of mountainous areas were mainly concentrated at a value of about $-0.03 \cdot 10 \text{ y}^{-1}$. The number of grid cells with a decreasing trend on plains, hills, and platforms were fewer, and their slopes were mainly concentrated around $-0.02 \cdot 10 \text{ y}^{-1}$ (Figure 5a). In mountainous areas, a large proportion of NDVI grid cells showed an increasing trend, but their slope values were generally low; in contrast, the slopes were mostly high in decreasing-trend grid cells. Therefore, the growth of NDVI of mountainous areas is slower than that in hilly or platform areas.

Among the nine vegetation types, the slopes of increasing NDVI of desert and cultivated vegetation areas were mainly concentrated around a value of $0.04 \cdot 10 \text{ y}^{-1}$; in alpine vegetation, broadleaf forest, grassland, and other vegetation areas it was $0.03 \cdot 10 \text{ y}^{-1}$; in coniferous forest, shrub, and meadow areas it was $0.02 \cdot 10 \text{ y}^{-1}$ (Figure 5b). Except for cultivated vegetation, broadleaved forest, and shrub vegetation, the slopes of declining NDVI of alpine vegetation and meadow areas were the greatest, being mostly $-0.03 \cdot 10 \text{ y}^{-1}$, while those in coniferous forest and desert were mainly $-0.02 \cdot 10 \text{ y}^{-1}$ (Figure 5b).

Data above showed that the slope of increasing NDVI was highest for meadow vegetation, and the increasing slope in desert vegetation was higher than the other types. However, the slope was low in increasing grid cells of meadow vegetation during the same period.

### 3.3. Analysis of NDVI Abrupt-Change Year

In this paper, we analyzed the years when NDVI changed abruptly in the grid cells featuring a unimodal or U-shaped trend, in the Altay region. In general, such years for the former were mainly concentrated in the year 2000; those for the latter trend were concentrated in 2003 (Figure 6a).

The frequency of distribution for the abrupt-change year was clearly different among the four geomorphic types. In both plain and hilly areas, the abrupt-change year in the NDVI grid cells of the unimodal trend was mainly concentrated in 2000, which was consistent with that across the Altay region. The main abrupt-change years in mountains were 1993 and 1994, and in platforms, 1986 (Figure 6b). In both plain and platform areas, the abrupt-change year in the NDVI grid cells having a U-shaped trend was mainly concentrated in 2003, which was consistent with the whole Altay region. In the hilly areas,

the distribution of grid cells abrupt-change years of the U-shaped trend was scattered, and there was no clear aggregation year (Figure 6c).

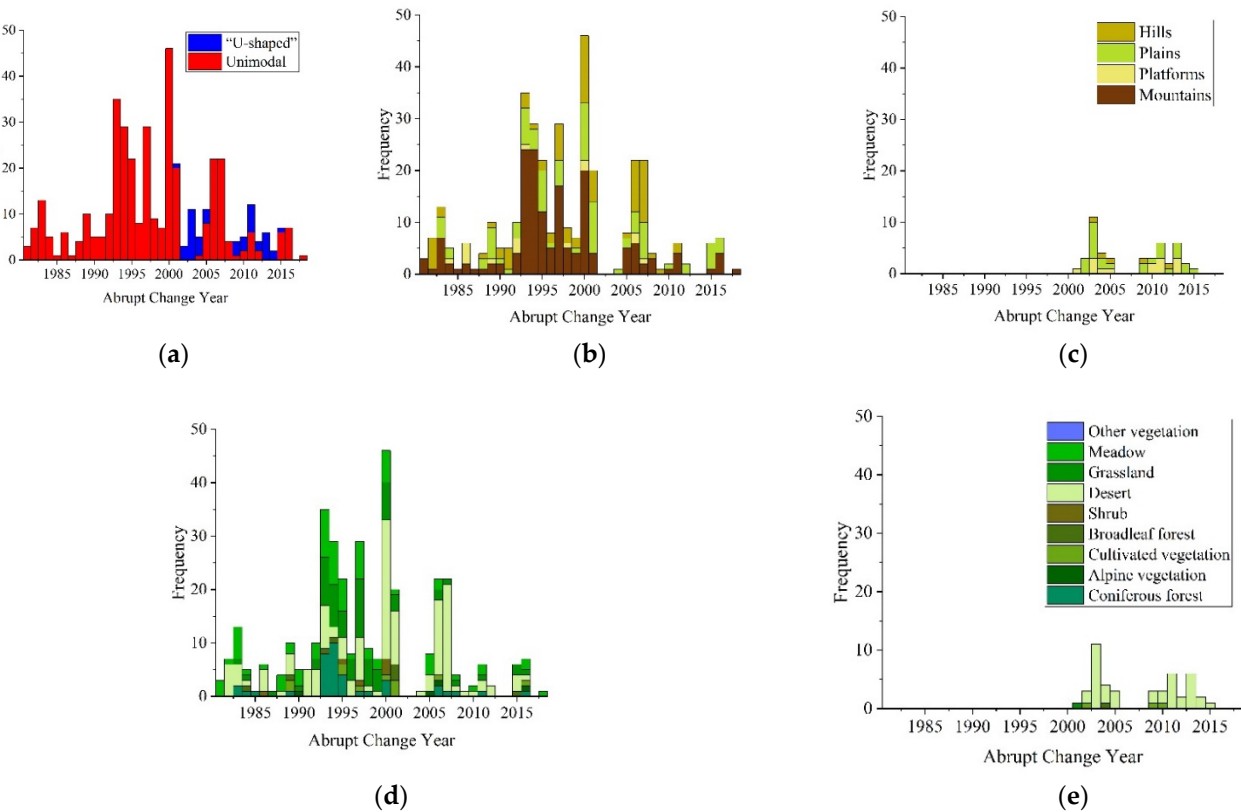

**Figure 6.** Histograms of the abrupt-change year of the NDVI ofof the Altay region from 1981 to 2018 for (**a**) both trends in the NDVI; for geomorphic types showing (**b**) unimodal trend and (**c**) U-shaped trend in the NDVI; for vegetation types showing (**d**) unimodal trend and (**e**) U-shaped trend in the NDVI.

The major abrupt-change years of the NDVI showed great differences among vegetation types. The major abrupt-change year of alpine vegetation was 2005; for coniferous forest it was 1994; for steppe it was 1997, for meadow it was 1993; and for cultivated vegetation it was 2001. Only shrub and desert vegetation types were consistent with the whole of the Altay region in having abrupt-change years concentrated in 2000 (Figure 6d). The U-shaped grid cells of the NDVI were mainly distributed in the desert vegetation, of which the abrupt-change year most commonly occurred in 2003. In broadleaf forest, grassland, desert, and cultivated vegetation, a small number of U-shaped grid cells were found, and although the distribution of their abrupt-change years was scattered, they were all concentrated around 2003 (Figure 6e).

*3.4. Influence of Climatic Factors on NDVI Dynamic*

Natural factors play a vital role in the growth of vegetation. Given that changes in water and heat will directly affect it, analyzing trends in temperature and precipitation can provide a basis for understanding vegetation growth dynamics. More than 50% of the 4892 grid cells in the Altay region underwent warming or humidification during 1981–2018. Furthermore, no grid cells of total annual precipitation showed a significant decreasing trend (Table 4). Therefore, climatic characteristics of the Altay region in the past 38 years were distinguished by significant "warming and wetting" changes.

**Table 4.** Trends of climatic factors from 1981 to 2018.

| Climatic Factors | Significant Increase | Significant Decrease | Unimodal | "U-Shaped" | Non-Significant |
|---|---|---|---|---|---|
| Annual mean temperature | 2744 (56.0%) | 29 (0.5%) | 8 (0.1%) | 864 (17.6%) | 1247 (25.4%) |
| Total annual precipitation | 3160 (64.5%) | 0 (0%) | 66 (1.3%) | 60 (1.2%) | 1606 (32.8%) |

Values are the number of grid cells (proportion) of the annual average temperature and total annual precipitation of different change characteristics.

The annual mean temperature increased significantly, occurring at 56.0% of the total number of grid cells concentrated in the central plain and platform areas. The rate of rising temperature declined from the interior to the periphery of the Altay region. The grid cells with non-significant changes in annual mean temperature were mainly distributed in the northern part of the study area. There, the landform is largely mountainous, with high-elevation vegetation types found, namely meadow, grassland, coniferous forest, and alpine vegetation. The U-shaped variation trend accounted for 17.6% of the total number of grid cells concentrated in the southwestern hilly desert area of the Altay region (Figures 1c and 7a).

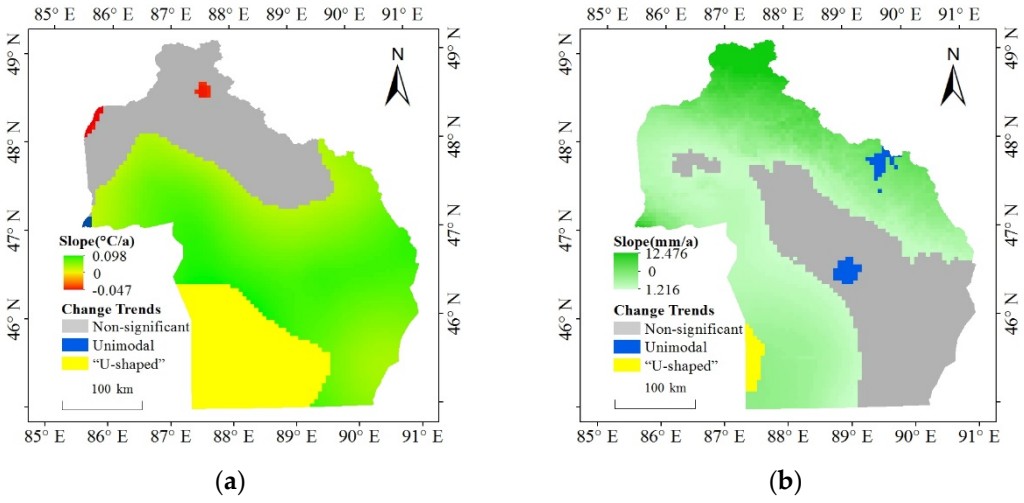

**Figure 7.** Spatial distribution of changes in (**a**) annual mean temperature and (**b**) total annual precipitation in the Altay region, from 1981 to 2018.

The total annual precipitation in the Altay region was dominated by a significantly increasing trend, this characterizing 64.5% of the total number of grid cells, and mainly distributed in the mountainous area in the northeast and the hilly area in the southwest of the Altay region; the highest rate of precipitation increase happened in the northern mountains. Those grid cells featuring a U-shaped trend accounted for 1.2% of the total number of grid cells distributed in the southwestern hilly area, and surrounded by grid cells with a significant increasing trend. The interspersed grid cells of the unimodal trend were distributed in the central region with non-significant trends and the northeast region with significant increase, together accounting for 1.3% of all grid cells. There was no significant decreasing trend for total annual precipitation in the Altay region. Grid cells in which non-significant trends occurred were widely dispersed in the central plain, platform and parts of the southeastern hilly area with low annual rainfall (Figures 1c and 7b).

The Pearson correlation analysis of the NDVI with climatic factors at the grid-cell scale revealed that total annual precipitation has a more extensive augmentative effect on NDVI than annual mean temperature in the Altay region. The NDVI often had an insignificant negative correlation with the mean annual temperature, these cases accounting for 45.95% of the total number of grid cells. However, among the grid cells having a significant

correlation, more had a significantly positive than negative correlation with annual mean temperature, accounting for 7.5% (Table 5). The largest number of correlations between NDVI and total annual precipitation were significantly positive, corresponding to 84.1% of all grid cells in the Altay region. And no grid cells had an insignificant negative correlation with total annual precipitation (Table 5).

**Table 5.** Correlation between the NDVI and climatic factors, from 1981 to 2018.

| Climatic Factors | Significant Positive Correlation | Significant Negative Correlation | Insignificant Positive Correlation | Insignificant Negative Correlation |
| --- | --- | --- | --- | --- |
| Annual mean temperature | 368 (7.5%) | 109 (2.2%) | 2166 (44.2%) | 2249 (45.95%) |
| Total annual precipitation | 276 (5.6%) | 501 (10.2%) | 4115 (84.1%) | 0 (0%) |

Values are the number of grid cells (proportion) for correlations between the NDVI and the annual average temperature or total annual precipitation ($p < 0.05$).

Spatially, the Pearson correlation coefficient between the NDVI and the annual mean temperature gradually decreased from the central part of the Altay region to its surrounding area. Among them, the grid cells having a positive correlation were found mostly distributed in the river and lake basins in the central part of the study area, as well as some mountain valleys. The grid cells with a negative correlation are mainly distributed in the desert, in the region's northwest and southeast. The correlation between the NDVI and the annual mean temperature in other vast parts was low and their distribution characteristics are not significant (Figures 1c and 8a). The grid cells that had a positive correlation between NDVI and total annual precipitation were mainly located in the central and northern areas of the Altay region. The increase of precipitation in this area would have promoted the growth of vegetation. The correlation between the NDVI and total annual precipitation was negative only in the northern mountain highlands, the western Urungu Lake and parts of the southeastern hills area. Finally, the correlation between the NDVI and total annual precipitation was low in the northwestern and southern desert areas, as well as at the foot of the Altay Mountains (Figures 1c and 8b).

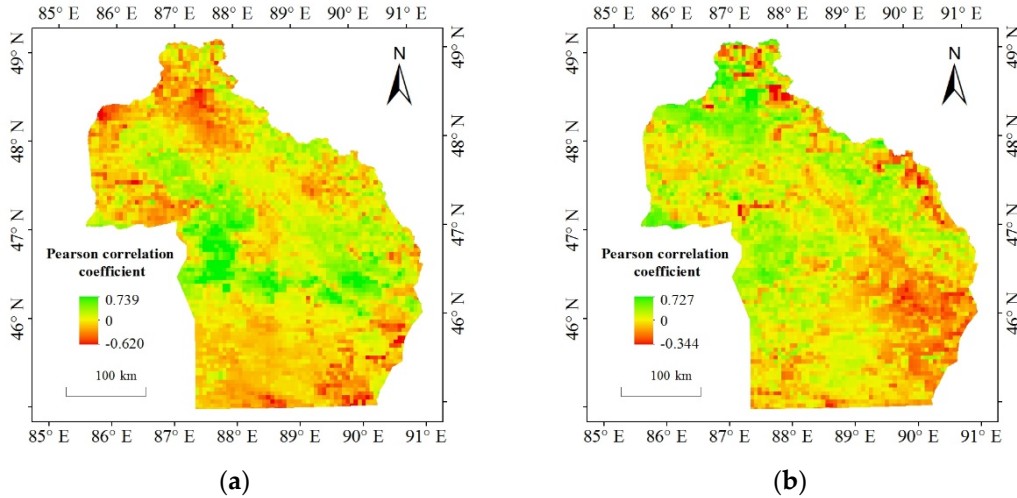

    (**a**)                                                 (**b**)

**Figure 8.** Spatial distribution for correlations of the NDVI with annual mean temperature and precipitation in the Altay region from 1981 to 2018: (**a**) between the NDVI and annual mean temperature, (**b**) between NDVI and total annual precipitation.

There were insignificant positive correlations between annual mean temperature and the NDVI of the plain, platform, and mountain areas. The hilly area showed an insignificant negative correlation, with more than 50% of the total grid cells of this geomorphic type

(Table 6). Among the grid cells with significant correlation between different geomorphic types, the significant positive correlation dominated, accounting for 1.7% (mountain) to 15.8% (platform) of the total grid cells per geomorphic type (Table 6). This analysis showed that the overall NDVI of the platform area was higher, and its vegetation coverage was better. The rising annual mean temperature in this area had the strongest effect on the NDVI. In desert or hilly areas with sparsely distributed vegetation, evapotranspiration may be intensified due to rising temperature, thus aggravating water stress and inhibiting the growth of the NDVI.

**Table 6.** Correlations between the NDVI and climatic factors for different landform types.

| Climatic Factors | Geomorphic Type | Significant Positive Correlation | Significant Negative Correlation | Insignificant Positive Correlation | Insignificant Negative Correlation |
|---|---|---|---|---|---|
| Annual mean temperature | Plains | 175 (13.2%) | 27 (2.0%) | 614 (45.5%) | 504 (38.1%) |
| | Platforms | 79 (15.8%) | 4 (0.8%) | 247 (49.4%) | 169 (33.8%) |
| | Hills | 86 (5.7%) | 51 (3.4%) | 540 (36.2%) | 813 (54.5%) |
| | Mountains | 28 (1.7%) | 27 (1.7%) | 765 (48.3%) | 763 (48.1%) |
| Total annual precipitation | Plains | 75 (5.6%) | 107 (8.1%) | 1138 (86.2%) | 0 (0%) |
| | Platforms | 29 (5.8%) | 28 (5.6%) | 442 (88.5%) | 0 (0%) |
| | Hills | 100 (6.7%) | 183 (12.2%) | 1207 (81.0%) | 0 (0%) |
| | Mountains | 72 (4.5%) | 183 (11.5%) | 1328 (83.8%) | 0 (0%) |

Values are the number of grid cells (proportion) for correlations between the NDVI and annual average temperature or total annual precipitation ($p < 0.05$) in each landform type.

In the four geomorphic types, an insignificant positive correlation between the NDVI and total annual precipitation was common, accounting for 81.0% (hills) to 88.5% (platform) of each geomorphic type grid cells (Table 6). No grid cells in the NDVI of the Altay region and total annual precipitation had a significant negative correlation (Table 6). Therefore, the NDVI mainly increased with more precipitation in each geomorphic type in the Altay region. This promotion effect was strongest for platform and plain areas. Compared with other geomorphic areas, the proportion of grid cells with significant positive correlation between total annual precipitation and the NDVI was lower in the mountainous area.

The correlations between the NDVI and annual mean temperature were not the same among vegetation types. Among the types with significant positive correlations between the NDVI and annual mean temperature, the number of grid cells in cultivated vegetation and broadleaved forest was the largest, respectively, accounting for 34.1% and 28.3% of their total grid cells. In the alpine vegetation, cultivated vegetation, broadleaved forest, grassland and meadow area NDVI was mainly not significantly (positively) correlated with annual mean temperature, representing 56.9%, 46.1%, 46.9%, 52.0%, and 48.7% of the vegetation type grid cells, respectively. For coniferous forest, shrub, desert and other vegetation types, there were mainly insignificant negative correlations, accounting for 47.5% (desert) to 65.2% (shrub) of the total number of grid cells per vegetation type (Table 7). Thus, with a rising annual mean temperature, the NDVI of cultivated vegetation and broad-leaved forest increased more clearly than other vegetation types. On the contrary, the decrease in temperature will promote the growth of coniferous forest, shrub, desert slightly. Rising global temperature may inhibit their growth in the future.

We found insignificant positive correlations between NDVI and total annual precipitation in all kinds of vegetation types, accounting for 69.8% (alpine vegetation) to 96.5% (cultivated vegetation) of the total number of grid cells for each vegetation type (Table 7). The increase in total annual precipitation likely promoted the growth of vegetation in all vegetation types, especially coniferous forest. However, in alpine, broadleaf forest, meadow, desert and other vegetation distribution areas, the presumed promotion effect was weak.

**Table 7.** Correlations between the NDVI and climatic factors for different vegetation types.

| Climatic Factors | Vegetation Type | Significant Positive Correlation | Significant Negative Correlation | Insignificant Positive Correlation | Insignificant Negative Correlation |
|---|---|---|---|---|---|
| Annual mean temperature | Coniferous forest | 4 (1.7%) | 9 (3.9%) | 95 (42.0%) | 118 (52.2%) |
| | Alpine vegetation | 2 (1.0%) | 2 (1.0%) | 106 (56.9%) | 76 (40.8%) |
| | Cultivated vegetation | 40 (34.1%) | 0 (0%) | 54 (46.1%) | 23 (19.6%) |
| | Broadleaf forest | 23 (28.3%) | 0 (0%) | 38 (46.9%) | 20 (24.6%) |
| | Shrub | 0 (0%) | 0 (0%) | 8 (34.7%) | 15 (65.2% |
| | Desert | 252 (8.7%) | 81 (2.8%) | 1173 (40.8%) | 1368 (47.5%) |
| | Grassland | 20 (3.1%) | 3 (0.4%) | 336 (52.0%) | 286 (44.3%) |
| | Meadow | 21 (3.0%) | 14 (2.0%) | 334 (48.7%) | 316 (46.1%) |
| | Other vegetation | 6 (10.9%) | 0 (0%) | 22 (40.0%) | 27 (49.0%) |
| Total annual precipitation | Coniferous forest | 7 (3.0%) | 10 (4.4%) | 209 (92.4%) | 0 (0%) |
| | Alpine vegetation | 11 (5.9%) | 45 (24.1%) | 130 (69.8%) | 0 (0%) |
| | Cultivated vegetation | 3 (2.5%) | 1 (0.8%) | 113 (96.5%) | 0 (0%) |
| | Broadleaf forest | 9 (11.1%) | 11 (13.5%) | 61 (75.3%) | 0 (0%) |
| | Shrub | 1 (4.3%) | 1 (4.3%) | 21 (91.3%) | 0 (0%) |
| | Desert | 189 (6.5%) | 314 (10.9%) | 2371 (82.4%) | 0 (0%) |
| | Grassland | 21 (3.2%) | 17 (2.6%) | 607 (94.1%) | 0 (0%) |
| | Meadow | 34 (4.9%) | 88 (12.8%) | 563 (82.1%) | 0 (0%) |
| | Other vegetation | 1 (1.8%) | 14 (25.4%) | 40 (72.7%) | 0 (0%) |

Values are the number of grid cells (proportion) for correlations between NDVI and annual average temperature or total annual precipitation ($p < 0.05$) in each vegetation type.

## 4. Discussion

The results revealed that most grid cells in the NDVI of the Altay region were marked by a non-significant trend of change during the 1981–2018 study period, while a significantly increasing trend was a secondary feature of this region. The spatial distribution characteristics of NDVI trends are generally consistent with the findings of Liu et al. [52]. Among different geomorphic types, the NDVI of plain, platform, and hilly areas mainly displayed non-significant changes. In the mountainous area, the NDVI ofcreased significantly, especially in valleys, depressions and riverbanks, albeit slowly, over time. This result is consistent with findings of Xiong et al. [53]. The principal reason for this phenomenon in the mountainous area may be that its snowmelt supplies more water to vegetation along the runoff banks. That NDVI has a low correlation with total annual precipitation, but a high correlation with annual mean temperature, further proves this phenomenon. He et al. [54] found that cultivated vegetation, grassland, and desert vegetation increased the most in northern China during 1982–2015, which is similar to the results of our study. In addition, the NDVI rate of increase is pronounced in coniferous forest and broad-leaved forest in the Altay region, whereas its change in desert vegetation is small because of sparse vegetation there. Only the grid cells close to the water source show rapid greening over time [55]. Under the influence of human activities, the range and rate of NDVI growth of cultivated vegetation are increasing, which is consistent with prior research [56]. The overall NDVI growth trend of coniferous forest, broadleaved forest, grassland, and meadow vegetation was relatively slow, in line with findings reported by Ma et al. [57].

In our study area, there was an insignificant correlation between the NDVI and annual mean temperature, and this relationship differed significantly among vegetation types. Piao et al. [58] and Zhu et al. [59] also obtained similar results. They also found that the contribution of warming to vegetation growth in the northern hemisphere is gradually diminishing. This phenomenon is related to the drought trend in temperate regions of the Northern Hemisphere, and the adaptation of vegetation growth to climate warming at high latitudes [60,61]. However, Kong et al. [62] did not reach the same view, a discrepancy, perhaps related to the different scales used between the studies. Among geomorphic types,

in the plain, platforms, and mountainous areas, the NDVI and annual mean temperature have predominantly insignificant positive correlations, while those grid cells with insignificant negative correlations are mostly limited to hilly parts. This phenomenon could be due to the relatively high elevation of the Altay region, where the accumulated temperature of its vegetation is relatively low [63]. The annual mean temperature captures the mean values of the four seasons on an annual scale. In mountainous and hilly areas at high elevation, the annual mean temperature captures spring and autumn temperature information to a certain extent [64]. However, the vegetation growth phenology in these areas is short and concentrated in summer [65]. Therefore, the correlation between average annual temperature and the NDVI is not significant in high-elevation, mountainous and hilly areas [58]. The hilly areas consist mostly of desert, where plants are sparse; hence, it is difficult for climate change to significantly impact the vegetation in that geomorphic type in only a few decades. Further, because of already high temperatures, as they continue to rise, evaporation will increase, which will affect the growth of desert vegetation, thus leading to its prevailing insignificant negative correlation between the NDVI and annual mean temperature [66]. In addition, cultivated vegetation and broadleaf forest are mainly distributed along rivers. Precipitation is not the limiting factor for vegetation growth in these areas because of river recharge and melting water of alpine ice and snow caused by the temperature rise. Therefore, a significant positive correlation between the annual mean temperature and the NDVI characterizes the cultivated vegetation and broad-leaved forest. In the higher-elevation vegetation, distribution areas, such as alpine vegetation, grassland, and meadow, their NDVI and annual mean temperature mainly showed an insignificant positive correlation [67,68]. While coniferous forest and shrub vegetation types changed differently under the influence of annual mean temperature, as found in previous research, their NDVI was mainly negatively correlated with annual mean temperature [69].

The Altay region has a temperate continental climate and is an arid zone, where water is the main factor limiting vegetation growth [70]. Therefore, most of the study area is positively correlated with total annual precipitation, especially in plains and platform areas, where there are no rivers or lakes. Lacking another water supply, vegetation in the region strongly relies on precipitation, so there is a significant positive correlation between the two. Sun et al. [71] also obtained similar results. Vegetation growing in mountain and river valleys has access to abundant water and can rely less on precipitation, leading to its low correlation with total annual precipitation [72]. Alpine vegetation and desert vegetation have non-significant changes due to their sparse plants; hence, most of their grid cell NDVIs show an insignificant positive correlation with total annual precipitation [73].

Since the start of the 21st century, climate change has driven an increase in precipitation in the Altay region, where its climate is becoming more suitable for vegetation growth. Further, government policies, such as zoning and rotation grazing, resting grazing and banning grazing, as well as the *Regulations on Ecological and Environmental Protection in Altay Region*, not only promote the environmental and ecological protection of the Altay region and the sustainable development of its local economy, but also effectively protect local vegetation and promote greater NDVI [74]. Accordingly, NDVI is not only affected by climatic factors, but also human activities that are important factors affecting vegetation change in the Altay region.

In general, the NDVI of the Altay region has been developing in a good direction in the recent 38-year period studied here. Nevertheless, under the influence of climate change and human activities, there is still a "mosaic" of declining NDVI trends in the study area. This poses a hidden danger to the sustainable environment and the stability of agriculture and animal husbandry in the region [75]. Therefore, it is imperative to put forward targeted guidance on policy, to improve the vegetation greening situation in the Altay region under the background of global climate change, and promote local sustainable development.

This paper emphasizes the analysis of the spatio–temporal changes of a long time series of NDVIs (38 years) and their driving force in terms of climatic indicators. Although there are many datasets with a higher resolution, due to the long time series of NDVI

data (NOAA CDR AVHRR NDVI V5) in this study, its rough spatial resolution was an unavoidable compromise. In the future, it is necessary to use higher resolution NDVI datasets or appropriate methods to improve the resolution of data sets for further analysis. Specifically, this study aims at elucidating the relationship of annual maximum NDVI to annual mean temperature and total annual precipitation. However, the CMFD dataset in this study still has the problem of a low resolution. However, it was chosen because of its applicability in our study area. The low-resolution problem of CMFD was solved by GWR downscaling. If there is a long-term series of high-resolution datasets supported by more site data in this study area in the future, the influencing factors of the vegetation change trend of complex underlying surfaces can be better analyzed. Therefore, although these relationships can capture the average feedback state between vegetation and climatic factors on the complex underlying surface in arid areas at high elevation and latitudes, there are still many problems to be solved. In the near future, fine-scale temporal studies using seasonal and monthly analytical scales will also be the next step.

## 5. Conclusions

(1) During 1981–2018, the annual NDVI of the Altay region showed a non-significant trend, followed by a significant increase trend. Among the geomorphic types, the northern mountain has the highest grid cell proportion, with a significant increasing trend. Among its nine vegetation types, the main trend in the NDVI of coniferous forest, cultivated vegetation, broadleaf forest, grassland, and meadow was of a significant increase over time.

(2) For grid cells with a linear temporal trend for the NDVI, the proportion of those with a significant increase in the NDVI was considerably higher than that with a significant decrease trend, which applied to all geomorphic types and vegetation types. Statistical analysis of the rate of change in the NDVI showed that it expanded fastest in hilly and platform areas, and receded fastest in the mountainous area. Among vegetation types, desert vegetation and cultivated vegetation grew fastest, while alpine vegetation and meadow declined at the fastest rate.

(3) Statistical analysis of grid cells with nonlinear changes showed that, in the study period, the abrupt-change year of annual NDVI with a unimodal trend was mainly concentrated in 2000, and a U-shaped trend was mainly concentrated in 2003. Among geomorphic or vegetation types, there were some differences in their abrupt-change year. The main year that the platform area showed a unimodal trend was 1986, but it was much later for the mountain areas (1993 and 1994). The U-shaped grid cell abrupt-change years in the hilly area were scattered, precluding an obvious aggregation year, while there were no U-shaped grid cells in the mountainous area. The main abrupt-change years of unimodal grid cells for vegetation types were 2005 for alpine vegetation, and likewise, 1994 for coniferous forest, 1997 for grassland, 1993 for meadow, and 2001 for cultivated vegetation. The U-shaped grid cells were mainly distributed in desert vegetation, and the main abrupt-change years were consistent with the whole Altay region.

(4) The time series analysis of annual average temperature and total annual precipitation in the 38-year period shows that the whole Altay region presents a phenomenon of "warming and wetting". The correlation analysis showed that this region's NDVI was positively correlated with annual mean temperature and precipitation. Yet, among all geomorphic types, due to desert conditions in the hilly area, evapotranspiration is intensified due to rising temperature, rendering a higher proportion of grid cells with an insignificant negative correlation between the NDVI and annual mean temperature. Among vegetation types, the proportion of grid cells with an insignificant negative correlation between the NDVI and annual mean temperature was higher in coniferous forest, shrub, and desert vegetation. In the geomorphic or vegetation types, the NDVI was mainly positively correlated with total annual precipitation. This indicates that precipitation has a more significant augmentative effect on vegetation in the Altay region.

**Author Contributions:** All authors contributed to the design and development of this manuscript. Conceptualization, J.C. and H.W.; methodology, J.C.; software, Y.L.; validation, Y.Y. and J.F.; formal

analysis, H.W.; investigation, Y.Y. and J.F.; resources, Y.L.; data curation, J.C., Y.L. and Y.Y.; writing—original draft preparation, Y.Y.; writing—review and editing, J.C.; visualization, Y.L.; supervision, J.C.; project administration, H.W.; funding acquisition, J.C. All authors have read and agreed to the published version of the manuscript.

**Funding:** This research was supported by the National Natural Science Foundation of China (32260280).

**Institutional Review Board Statement:** "Not applicable" for studies not involving humans or animals.

**Informed Consent Statement:** Not applicable.

**Data Availability Statement:** The data that support the findings of this study are available from the corresponding author upon reasonable request.

**Acknowledgments:** We thank the College of Resources and Environment of Xinjiang Agricultural University for their support.

**Conflicts of Interest:** The authors declare no conflict of interest.

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
