# Peer review of "Characteristics of NDVI Changes in the Altay Region from 1981 to 2018 and Their Relationship to Climatic Factors"

_land, doi:10.3390/land12030564_

Round 1
Reviewer 1 Report
The study by Yang et al. examines the spatial and temporal variations in NDVI across the Altay region in China. The authors examine NOAA CDR AVHRR NDVI V5 data across a 38 year span, exploring the correlation of the pattern of NDVI change with the variation in climatic factors and vegetation type.
Overall, the manuscript tackles a very difficult goal, namely, understanding the sources of variation in vegetation vigour as a proxy of productivity, across a very complex and diverse (both geomorphological and in terms of vegetation) region. The authors provide a clear concise description of their goals and the methodological approach taken. However, I have some concerns regarding the data used and the analytical approach.
First, I have concerns regarding identification of the pattern of NDVI change. Data in each of the study area pixels have been fitted with linear models, attempting to identify whether a linear or non linear model is the best fit. While the authors describe a sequential model fitting protocol, model selection should actually follow the guidelines indicated by Peng et al (2011) (ref 28 in the manuscript) or Burhnam and Anderson (2003), examining not only the individual AIC values, but also calculating AIC differences as Δ?=????−?????? and ensuring that the absolute Δ? is at least 2. In addition to this, it would be relevant to examine what is the coefficient of determination for each of the selected models. This is relevant, as a significant model (even a linear one) may account for a low proportion of observed variation in the response (dependent) variable. I would suggest that the authors examine the range of R2 values and report in a supplementary material file or figure. In addition to these concerns, it would be very helpful to be able to examine some examples of the time series patterns selected by the model selection procedure. This is particularly relevant for nonlinear models which show maxima or minima quite close to the start or end of the study period (see Figure 3). It may well be that unimodal or U-shaped patterns with inflection points prior to 1985 or after 2010 may be artifactual. This could be cleared by showing either information on the R2 of the fitted models or by showing representative time series. Also, it is not stated in the methods section whether the NDVI dataset is annual mean, as opposed to temperature and precipitation which are annual values. If so, it would be interesting to examine also the variability around the annual means, as these may provide additional insights.
Despite these reservations, the described pattern is quite interesting, and encompasses a large amount of work. I am curious as to what is the spatial pattern of the models fitted to NDVI. The authors report the spatial pattern of different trends or dynamic signatures climatic variables (see Figure 4), but this has not been done for NDVI. Instead, the authors provide a detailed qualitative frequency analysis of the association of different dynamic patterns with Geomorphic and Vegetation types in tables 1 and 2. While the qualitative analysis is informative, I would suggest the frequency tables are analysed with some measures of independence (Chi square test) or association (Fisher's exact test). These would provide statistical arguments as to whether there is association or not. However, regardless of the analytical strategy, the main sources of variation seem to be associated with associated with mountains, as well as with some vegetation types. However, while Tables 1 and 2 indicate that there are large proportions of cells with non significant patterns, this is not reflected in the histograms shown in Figure 2. Specifically, while tables 1 and 2 show large percentages of non significant dynamics (39.6% to 68.1% for geomorphic types and 12.8% to 69.7% for vegetation dynamics) All these should have a slope of zero. However, histograms shown in Figure 2 report no zero slope values.I would encourage the authors to provide further details as to why this discrepancy arises.
A third and final concern has to do with the correlation analysis between NDVI and climatic variables. In this regard, the study uses information obtained at different spatial resolutions. While NDVI data have a 5 km resolution, climatic data sets have a 10km resolution. The authors indicate that climatic data have been resampled to a higher resolution using nearest neighbor method. While this has the advantage of providing a simple method that preserves data values, it has the disadvantages of leading to the loss of spatial resolution, introducing spatial artifacts and resulting in a poor representation of spatial features. In fact, the method does not increase the spatial resolution of the data, and may actually result in a loss of resolution and a smoothing of the data. In addition, it can create artifacts such as blocks of the same values in the resampled data, especially if the original data contains sharp transitions in values. This introduces spatial autocorrelation as a result of the resampling technique. Further, this method may not adequately represent complex spatial features in the data, such as ridges or valleys, as the resampled data is determined only by the values of the closest original data points. This is particularly relevant, given the study area and its complex geomorphology. In addition, the authors indicate that the data for geomormphic regionalization data and vegetation regionalization are at a different scale (1:1000000) than the AVHHR and CMFD data. Hence, it would be relevant to indicate what is the minimum detectable size or pixel resolution in order to assess whether these two latter data sets are in fact resampled or upscaled to match the 5km NDVI data. As mentioned earlier, while the authors show the spatial pattern of dynamic types for temperature and precipitation ( increase, decrease, unimodal, u-shaped and constant) these are not in fact correlated with the corresponding spatial pattern of dynamic types for NDVI ( increase, decrease, unimodal, u-shaped and constant). Actually, it seems that the correlation has been calculated with the raw time series, as suggested by the use of the Pearson correlation coefficient. In this regard, this assumes there are no lags in the association, and a time series approach would seem adequate unless the authors discuss why no lags would have to be taken into consideration.
Author Response
Response to Reviewer 1 Comments
Dear reviewer:
Thanks a lot for having reviewed our manuscript. Now we have been extensively revised the manuscript according to the reviewers' comments. Most of the revisions are in the manuscript. Some explanations regarding the revisions of our manuscript are as follows.(The image cannot be displayed on the web page. Please see the word.docx for details)
Point 1: First, I have concerns regarding identification of the pattern of NDVI change. Data in each of the study area pixels have been fitted with linear models, attempting to identify whether a linear or non linear model is the best fit. While the authors describe a sequential model fitting protocol, model selection should actually follow the guidelines indicated by Peng et al (2011) (ref 28 in the manuscript) or Burhnam and Anderson (2003), examining not only the individual AIC values, but also calculating AIC differences as Δ?=????−?????? and ensuring that the absolute Δ? is at least 2. In addition to this, it would be relevant to examine what is the coefficient of determination for each of the selected models. This is relevant, as a significant model (even a linear one) may account for a low proportion of observed variation in the response (dependent) variable. I would suggest that the authors examine the range of R2 values and report in a supplementary material file or figure. In addition to these concerns, it would be very helpful to be able to examine some examples of the time series patterns selected by the model selection procedure. This is particularly relevant for nonlinear models which show maxima or minima quite close to the start or end of the study period (see Figure 3). It may well be that unimodal or U-shaped patterns with inflection points prior to 1985 or after 2010 may be artifactual. This could be cleared by showing either information on the R2 of the fitted models or by showing representative time series. Also, it is not stated in the methods section whether the NDVI dataset is annual mean, as opposed to temperature and precipitation which are annual values. If so, it would be interesting to examine also the variability around the annual means, as these may provide additional insights.
Response 1: The method for identifying NDVI change patterns was added in detail in resubmitted manuscript. And in order to understand easily, we have make a special technical flowchart of our article. The 38-year NDVI change was fitted by linear model (Sen's slope) and nonlinear (Quadratic Function), respectively. The significance of the linear model(Pl) and the significance of the nonlinear model(Pn) was compared. If both Pl and Pn are greater than 0.05, both linear and nonlinear models reach the significance level, then determine whether it is a linear or nonlinear model by Akaike Information Criterion(AIC). The gridcell is a linear model while AIC of linear model(AICl) is less AIC of nonlinear model(AICn). On the contrary, this gridcell is a nonlinear model. If the linear model has a positive(negative) slope, then the gridcell shows a significant increasing(decreasing) trend. Correspondingly, If the slope of nonlinear model shows positive-negative(negative-positive) characteristic, the gridcell is a unimodel(“U-shaped”). NDVI gridcells with U-shaped and unimodel characteristic need to be analyzed about abrupt change year by Mann-Kendall and Sliding t test algorithms. The NDVI Dynamic was analyzed mainly by Pearson’s correlation coefficient algorithm .
Spatial pattern analysis of vegetation response to climate is based on different geomorphic and vegetation types. Here is the flow chart in the manuscript.
Figure 3. Technology flow chart.
AIC is a measure of the Goodness of fit (GOF) of a statistical model. The AICs of the two models can be directly compared if the following two conditions are satisfied. First, the comparing models fitted by maximum likelihood to the same data. Second, AICs of comparing models were obtained with a good fit. For a given spatially indexed gridcell, it fits linear and nonlinear models with the same 38 years of data. Therefore, the first condition was satisfied. AIC values were compared only when the significance of the fit was less than 0.05 for both linear and nonlinear models. Thus, both linear and nonlinear models which reached significance levels were good fitted. Therefore, the second condition was satisfied. In our article, AIC can be used to compare the goodness of fit of a linear model or a nonlinear model about NDVI and climate data.
Rather than the coefficient of determination(R2), we pay more attention to the probability(p) of models fitted. Because p value can reflect significance level, which is more statistically significant. Therefore, we choose p value as the criterion to distinguish the linear model from the nonlinear model.
In addition, the NDVI used in this study is the annual maximum, the precipitation is the total annual precipitation, the temperature is the annual average temperature. In the near future, fine-scale temporal studies using seasonal and monthly analytical scales will also be the next step.
Point 2: Despite these reservations, the described pattern is quite interesting, and encompasses a large amount of work. I am curious as to what is the spatial pattern of the models fitted to NDVI. The authors report the spatial pattern of different trends or dynamic signatures climatic variables (see Figure 4), but this has not been done for NDVI. Instead, the authors provide a detailed qualitative frequency analysis of the association of different dynamic patterns with Geomorphic and Vegetation types in tables 1 and 2. While the qualitative analysis is informative, I would suggest the frequency tables are analysed with some measures of independence (Chi square test) or association (Fisher's exact test). These would provide statistical arguments as to whether there is association or not. However, regardless of the analytical strategy, the main sources of variation seem to be associated with associated with mountains, as well as with some vegetation types. However, while Tables 1 and 2 indicate that there are large proportions of cells with non significant patterns, this is not reflected in the histograms shown in Figure 2. Specifically, while tables 1 and 2 show large percentages of non significant dynamics (39.6% to 68.1% for geomorphic types and 12.8% to 69.7% for vegetation dynamics) All these should have a slope of zero. However, histograms shown in Figure 2 report no zero slope values.I would encourage the authors to provide further details as to why this discrepancy arises.
Response 2: In the article for lack of space distribution of NDVI variation trends, we have added the Figure 4. to 3.1 Interannual Variation in NDVI.
Figure 4. Spatial distribution of changes of NDVI in the Altay region, from 1981 to 2018.
Through literature review, I learned that chi-square distribution is continuous distribution, and frequency can only appear in the form of integers, so the calculated statistics are discontinuous. At present, Chi-square test is mainly used to compare two rates or two components. The Chi-square test for the comparison of multiple rates or multiple components and the correlation analysis of classified data. And the Fisher's exact test is a statistical significance test used in the analysis of contingency tables. Thank you very much for the two testing methods you proposed. I believe that I will definitely provide statistical arguments as to whether the data association or not in my subsequent research.
That's exactly what you think - NDVI in the mountain and high elevation vegetation distribution places the most significant change, and is also more unique. Therefore, this paper also puts a lot of emphasis on the analysis of this region. The underlying surface of Altay region is complex and diverse, and there are many geomorphologic types and vegetation types distributed here, which also creates more research significance. However, in this paper, I prefer to elaborate the characteristics and distribution of NDVI with significant changes, because it is of more important significance for us to deal with climate change and maintain ecological balance in the future. Therefore, after obtaining different variation trends of NDVI in Altay region, I chose to prioritize the analysis of Significant increase, Significant decrease, Unimodal, "U-shaped", and Non-significant trends. And by analyzing the slope of the gridcell with a linear change trend, the area with a more drastic change was found. The year of NDVI abrupt change was obtained by detecting the abrupt change points of the gridcell with nonlinear trend change. These studies have more important guiding significance for the follow-up research and provide suggestions on environmental protection. The gridcells with a non-significant change trend were also potential ecological threats. The future also requires us to provide more effective methods to analyze the characteristics of the changes in the non-significant change gridcells and to predict future trends. Thanks again for your suggestion, although in this article reflect less, but this is my future plan to solve the problem. I hope your suggestions can also be reflected in my later research.
Point 3: A third and final concern has to do with the correlation analysis between NDVI and climatic variables. In this regard, the study uses information obtained at different spatial resolutions. While NDVI data have a 5 km resolution, climatic data sets have a 10km resolution. The authors indicate that climatic data have been resampled to a higher resolution using nearest neighbor method. While this has the advantage of providing a simple method that preserves data values, it has the disadvantages of leading to the loss of spatial resolution, introducing spatial artifacts and resulting in a poor representation of spatial features. In fact, the method does not increase the spatial resolution of the data, and may actually result in a loss of resolution and a smoothing of the data. In addition, it can create artifacts such as blocks of the same values in the resampled data, especially if the original data contains sharp transitions in values. This introduces spatial autocorrelation as a result of the resampling technique. Further, this method may not adequately represent complex spatial features in the data, such as ridges or valleys, as the resampled data is determined only by the values of the closest original data points. This is particularly relevant, given the study area and its complex geomorphology. In addition, the authors indicate that the data for geomormphic regionalization data and vegetation regionalization are at a different scale (1:1000000) than the AVHHR and CMFD data. Hence, it would be relevant to indicate what is the minimum detectable size or pixel resolution in order to assess whether these two latter data sets are in fact resampled or upscaled to match the 5km NDVI data. As mentioned earlier, while the authors show the spatial pattern of dynamic types for temperature and precipitation ( increase, decrease, unimodal, u-shaped and constant) these are not in fact correlated with the corresponding spatial pattern of dynamic types for NDVI ( increase, decrease, unimodal, u-shaped and constant). Actually, it seems that the correlation has been calculated with the raw time series, as suggested by the use of the Pearson correlation coefficient. In this regard, this assumes there are no lags in the association, and a time series approach would seem adequate unless the authors discuss why no lags would have to be taken into consideration.
Response 3: The first issue of different spatial resolution, I agree with your idea sincerely. In order to avoid uncertainties caused by the difference in resolution between NDVI and climate data, The Temperature and precipitation of CMFD were downscaled to 5 km×5 km spatial resolution by a geographically weighted regression(GWR) model. The results of the downscaled data were also analyzed in resubmitted manuscript. Since most of our study area belongs to high altitude mountains underlying surface, climate datasets contains large number of values of sharp transitions, such as ridges or valley. Thus, SRTM DEM was selected to downscale temperature and precipitation data by GWR. GWR based on DEM can avoid transition smoothing of data and obtain high spatial resolution meteorological data.
The resampling method used in the previous section does have the problem as you raised. In order to solve this problem, I have consulted some related references in recent days, and found that geographically weighted regression (GWR) is a relatively better method. In standard geographically weighted regression (GWR). The spatially varying - relationships between the dependent and each independent variable are explored under a constant and fixed scale, but for many processes their variation intensity may differ with respect to location and direction. In addition, the advantages and feasibility of this method were described in detail by Lu, B. et al. (2018), and further detailed supplementary explanations were made by Lu, B. et al. (2019) for the problems not clearly explained in the paper. This method has been recognized by people, and was widely used in improving the resolution of data products. Therefore, we used this method to re-process 0.1° resolution China Meteorological Forcing Dataset (CMFD), and modified the analysis results. The results show that although some data have changed, especially the correlation between NDVI and precipitation has concentrated to an insignificant positive correlation, which makes the results have a more explicit directivity. But the analysis found that changes in the data and results had less impact on the earlier conclusions. Of course, this method may not be the best choice, but for now, this method is indeed an effective way to solve the problem. And I am trying my best to find a better solution to use in future research.
Specifically, this study aims at elucidating the relationship of annual maximum NDVI to annual mean temperature and total annual precipitation. Thus,the rearch work was carried out on the annual scale. Numerous studies have shown that it is more appropriate to consider the lagged relationship of NDVI with temperature and precipitation on a monthly or seasonal scale. Thank you very much for your comments. Your suggestions and comments are very important and constructive for my next research. The lags between NDVI and climate data would be taken into consideration in the future.
Thanks again to the reviewer for the advice. I think they are very valuable references and can help improve the quality of the article. In response to the questions you raised, I have made a lot of revisions in the article and explained them above. If you have any other suggestions on this article, please let me know as soon as possible.

Reviewer 2 Report
Manuscript with ID: land-2217715 is very interesting and it surely has relevance to the Journal of Land. Authors paid a great care in preparation of the article. However, I would suggest looking into following recommendations
1. Authors are recommended to use flow diagram / flow chart to present the adopted methodology, allowing for better understanding among readers.
2. Numerous studies are available on utilizing a large-scale time series data to study variations in NDVI. In this sense the authors should clearly show the aspects in which their study is novel.
3. Throughout the manuscript, authors are using the phrase high elevation, but elevation range of the study area is not mentioned. Please rectify and use any up to date digital elevation model to describe ranges of elevation.
4. Resolution of most of the figures is questionable, as axis labels and grid labels on most of the plots and maps are not clear.
5. Some other aspects including merits, demerits and limitation of the datasets utilized in current study should be covered.
6. Spatial resolution of the utilized dataset is too course. Discuss any possibility to validate the accuracy of obtained NDVI and other climatic factors such as annual precipitation and annual mean temperature.
7. Please, enlarge and re-arrange all Figures and their font sizes to guide the reader properly in all corresponding sections.
8. Selection of the criteria to adopt the targeted/used dataset should be discussed.
9. Avoid references older than 5 years (>2017)
10. Consider adding the web links to the data used (in the reference list), for instance, NOAA CDR AVHRR NDVI V5, geomorphic and vegetation zoning data.
Author Response
Response to Reviewer 2 Comments
Dear reviewers:
Thanks a lot for having reviewed our manuscript. Now we have been extensively revised the manuscript according to the reviewers' comments. All revisions are in the manuscript of re-submission. The explanations about the revisions of our manuscript are as follows. (The image cannot be displayed in the web page.Please see word.doc for details.)
Point 1: Authors are recommended to use flow diagram / flow chart to present the adopted methodology, allowing for better understanding among readers.
Response 1: The suggestion to add flowcharts is very meaningful, I accept the comment. I have added flowcharts under 2.3. Methods of submitted manuscript. It introduces in detail the research methods of our article and the process of data analysis, so as to facilitate the reader's understanding.
Here is the flow chart in the manuscript.
Figure 3. Technology flow chart.
Point 2: Numerous studies are available on utilizing a large-scale time series data to study variations in NDVI. In this sense the authors should clearly show the aspects in which their study is novel.
Response 2: Thanks a lot for this point. I have added to this content in the Introduction, see the submitted manuscript. The following are some of the additions.
With the emergence of long time series NDVI dataset products, researches on various aspects have increased. The introduction of this paper mainly includes three parts about previous studies (the second paragraph of the introduction for details): Firstly, different NDVI datasets were used to analyze the variation trends of annual NDVI in different regions on the interannual scale. Secondly, the variation trends of NDVI in different regions are also different. Finally, there are differences in NDVI responses to the same meteorological factor in different study areas, and different responses to different meteorological factors in the same study area. According to the editor's suggestion, this paragraph has also been supplemented to make the cited article more logical and meaningful.
Point 3: Throughout the manuscript, authors are using the phrase high elevation, but elevation range of the study area is not mentioned. Please rectify and use any up to date digital elevation model to describe ranges of elevation.
Response 3: Regarding the lack of elevation range of the study area in the article, we chose DEM data of 90m (2.2.4. The Digital Elevation Model (DEM) for details). Moreover, fiddle and box charts were used to make statistics on the distribution frequencies of geomorphic types and vegetation types at different elevations (Figure 2. for details). Through Figure 2, we can not only get the elevation of the Altay region more intuitively, but also know the elevation distribution range of different geomorphic types and vegetation types.
The following are elevation distribution characteristics in geomorphic and vegetation types.
|
Elevation distribution of geomorphic types |
|
Elevation distribution of vegetation types |
Point 4: Resolution of most of the figures is questionable, as axis labels and grid labels on most of the plots and maps are not clear.
Response 4: To solve this problem, we have used software to appropriately improve the resolution of the pictures in the process of re-producing them to make them more clearer.
Point 5: Some other aspects including merits, demerits and limitation of the datasets utilized in current study should be covered.
Response 5: In section 2.2. Data Collection and Processing, we not only supplement the merits, demerits and limitation of all data, but also introduce data preprocessing in more detail. In addition, we have added references to fully justify the selection of these data.
Point 6: Spatial resolution of the utilized dataset is too course. Discuss any possibility to validate the accuracy of obtained NDVI and other climatic factors such as annual precipitation and annual mean temperature.
Response 6: Regarding the choice of datasets, in order to meet the trend of NDVI and its correlation with climate factors under the long time series studied in this paper. The time series length feature of NDVI data set is the most important basis for data selection. In this paper, we selected the mesoscale spatial resolution NOAA CDR AVHRR NDVI V5 dataset (the advantages and disadvantages of this dataset are detailed in the 2.2.1. NDVI Datasets section of the article). Although there are currently datasets with higher spatial resolution, the time series is short and cannot meet the requirements of this study. At present, this dataset is still widely used in the research of long time series NDVI. As a long-time series meteorological data set, CMFD is selected because it has better adaptability to the complex underlying surface in the arid region of northwest China and contains more data of meteorological stations than other meteorological datasets. There have been a number of validation efforts on this dataset, and the results have shown its value (2.2.2. Climate Datasets for details). In addition, the CMFD data were also downscaled to 5 km×5 km spatial resolution by a geographically weighted regression(GWR) model. The results of the downscaled data were also re-analyzed in submitted manuscript.
Point 7: Please, enlarge and re-arrange all Figures and their font sizes to guide the reader properly in all corresponding sections.
Response 7: During the revision process, we have enlarged and re-arranged all Figures and their font sizes to make all pictures clearer.
Point 8: Selection of the criteria to adopt the targeted/used dataset should be discussed.
Response 8: Sincerely thank you for your suggestion, we have supplemented this part in our resubmited manuscript. On the basis of high accuracy and applicability of the data, the time series length feature of NDVI dataset is the most important criteria for data selection. This avoided, as much as possible, that the captured trends are fluctuations of the NDVI/climate itself over a short perio. Based on this, NDVI and climate datasets with higher spatial resolution were preferentially selected.
The selection criteria of different data are supplemented in 2.2. Data Collection and Processing, and more references are cited for proof. In addition, data sources are clearly shown in Table 1. The selection criteria of different data are supplemented in 2.2. Data Collection and Processing, and more references are cited for proof. In addition, data sources are clearly shown in Table 1. In the last paragraph of the 4. Discussion, the reasons for the selection of data are further supplemented. In the future, it is also an urgent problem for us to use the updated high-resolution datasets of long time series or to solve the problem of poor resolution of existing datasets for research.
Point 9: Avoid references older than 5 years (>2017)
Response 9: The updated reference before 2017 was less than 14%.Unupdated references may not have been replaced because of their authority or because no more suitable new references were found.
Point 10: Consider adding the web links to the data used (in the reference list), for instance, NOAA CDR AVHRR NDVI V5, geomorphic and vegetation zoning data
Response 10: According to your suggestions, Table 1. has been added in 2.2. Data Collection and Processing to display data sources and the web links.
Thanks again to the editor for the advice. I think they are very valuable references and can help improve the quality of the article. In response to the questions you raised, I have made a lot of revisions in the article and explained them above. If you have any other suggestions on the article, please let me know as soon as possible.

Round 2
Reviewer 1 Report
This is my second review of the work by Yang Yan et al. The study examines the relationship between NDVI changes and different climatic factors. The study provides a detailed analysis of the distribution of NDVI trends, considering geomorphic and vegetation types, as well as the geographic distribution. The authors have addressed my main observations and comments, and I find the manuscript is now greatly improved. I have a few minor comments on typos, which I indicate below:
Throughout the manuscript I would suggest editing "gridcells" replacing it with "pixels" or either with "grid cells" throughout the text
line 25: I suggest to change "This study’s results of this study " to " This study’s results" deleting "of this study"
line 145: Add a space to "Administration[37,38]" so that it reads as "Administration [37,38]"
line 152: Edit the second capital letter "To this end, We " so that the sentence reads: " To this end, we".
lines 205 and 2007: I would suggest that "technology flow chart" be edited to "analysis workflow chart"
line 430: I suggest changing "insignificant" to "non significant"
Author Response
Dear reviewer,
First of all, thank you very much for your recognition of our first revisions result and your valuable advice again. We have completed the second revision with reference to your suggestions.The revised sections can be viewed in detail in the submitted manuscript.